# Tennis Serve Speed in Relation to Isokinetic Shoulder Strength, Height, and Segmental Body Mass in Junior Players

**DOI:** 10.3390/jfmk10010057

**Published:** 2025-02-05

**Authors:** Jan Vacek, Michal Vagner, Jan Malecek, Petr Stastny

**Affiliations:** 1Department of Sports Games, Faculty of Physical Education and Sport, Charles University in Prague, 162 52 Prague, Czech Republic; j.vacek.mail@gmail.com (J.V.); petr.stastny@ftvs.cuni.cz (P.S.); 2Department of Military Physical Education, Faculty of Physical Education and Sport, Charles University in Prague, 162 52 Prague, Czech Republic; jan.malecek@ftvs.cuni.cz

**Keywords:** tennis performance, isokinetic strength, grip strength, anthropometry

## Abstract

**Objectives:** The relationship between the isokinetic maximal strength of internal or external shoulder rotation and serve speed in tennis is well established, yet the influence of segmental mass, height, and high-speed shoulder rotation strength on serve performance in junior players remains unclear. This study aimed to investigate the relationship between concentric or eccentric isokinetic shoulder strength, segmental mass, height, and first-serve speed aimed at the T-target zone. **Methods:** Fifteen male junior competitive tennis players (mean ± SD: age 15.9 ± 0.9 years; height: 180.1 ± 7.2 cm; body mass: 66.1 ± 5.7 kg) were assessed for maximal isokinetic strength during concentric and eccentric internal and external shoulder rotations. Segmental mass (arm, leg, and trunk) was measured using dual-energy X-ray absorptiometry, and serve speed was recorded using a radar gun. **Results:** Concentric shoulder rotations at 210°/s demonstrated significant positive correlations with serve speed for both external (ρ = 0.71, *p* ≤ 0.01) and internal rotation (ρ = 0.61, *p* ≤ 0.05). Although lean arm mass partially mediated the relationship between shoulder strength and serve speed (indirect effect = 0.502, 95% CI: −0.156 to 1.145), this mediation effect was not statistically significant. Height was moderately correlated with serve speed (ρ = 0.68, *p* ≤ 0.01) but did not moderate the relationship between shoulder strength and serve speed. **Conclusions:** Concentric shoulder strength at higher angular velocities and segmental mass contribute to serve speed in junior tennis players. While height provides structural advantages, strength and lean mass play important roles, emphasizing the need for targeted training programs.

## 1. Introduction

The tennis serve is among tennis’s most technically and physically demanding strokes, requiring both refined technical proficiency and substantial physical strength and coordination across multiple body segments [1,2]. Serving effectively relies on the coordinated movement of multiple body segments (feet, lower limbs, trunk, shoulder, elbow, wrist, and hand), all contributing to an optimized energy transfer from the ground through the kinetic chain [3,4,5,6,7,8]. At the professional level, high-velocity serves are essential for competitive success, as they place opponents in defensive positions, thereby reducing their ability to return effectively [6,9,10,11]. Accordingly, understanding the key biomechanical and physical factors contributing to serve speed is crucial for performance improvement [12,13,14,15].

Among these physical factors, shoulder strength is pivotal for serve speed, particularly involving the internal rotators. Research consistently demonstrates that both the isometric and isokinetic strength of the shoulder’s internal and external rotators significantly influence racquet head speed, a primary factor underlying powerful serves [4,5]. Internal rotators substantially contribute to upper-arm angular velocity during the acceleration phase, affecting ball speed [5]. Additionally, both concentric and eccentric shoulder muscle actions are critical during both the acceleration and deceleration phases, facilitating racquet control and supporting injury prevention [16,17]. The relationship between shoulder strength and serve speed is well documented among elite players, where higher isometric and isokinetic strength correlates with faster serves [2]. However, findings in junior athletes are inconsistent. Some studies report significant correlations [9], while others find no association [4]. For example, one study found no significant correlation between maximum isokinetic shoulder strength and serve speed among nationally ranked players [3], while another identified muscular strength, power, and stiffness as crucial predictors of stroke velocity in junior players [9]. This variation suggests that junior players, still undergoing physical development, may rely on different performance determinants compared to professionals, highlighting the need for further exploration.

Furthermore, grip strength has also been identified as an important contributor to serve mechanics, influencing racquet control and force application during the serve [18,19]. Longer participation in tennis has been correlated with increased grip strength, highlighting the impact of sustained training on this attribute [20]. Studies examining the effects of resistance training on grip strength in tennis players aged 15–25 have shown significant grip strength improvements following targeted interventions [21]. Additionally, biomechanical analysis of the serve has identified key phases (loading, cocking, acceleration, contact, deceleration, and finish) in which strength and coordination play a critical role [6]. Lower body techniques during the loading phase (e.g., foot-back or foot-up styles) affect the vertical force initiation of the kinetic chain [4]. In the acceleration phase, the shoulder’s internal and trunk rotation primarily contribute to racquet head speed and serve velocity, reinforcing the importance of shoulder strength [5,22]. Furthermore, body segment masses (arm, leg, and trunk) affect force generation and transfer, with greater segmental mass enhancing energy transfer and racquet acceleration [23,24]. Understanding these interconnections is vital for refining training strategies in junior players.

Despite the well-established role of shoulder strength in serving speed among elite players, the specific impact of isokinetic concentric and eccentric shoulder strength training on junior players remains underexplored. This study includes the isokinetic testing of shoulder rotators at 210°/s and 300°/s to evaluate strength at high velocities in relation to serve speed. Additionally, grip strength, height, and body segment masses may provide valuable insights into serve mechanics and performance determinants. Therefore, this study aims to determine the correlations between serve speed and isokinetic concentric and eccentric shoulder strength, body mass, segmental masses, height, and grip strength in junior tennis players. We hypothesize that serve speed in junior tennis players positively correlates with the maximum isokinetic concentric strength of the shoulder’s internal and external rotators. Furthermore, we hypothesize that arm and trunk lean mass mediate this correlation, while height moderates it, with taller players exhibiting a stronger correlation between shoulder strength and serve speed.

## 2. Materials and Methods

### 2.1. Study Design

This cross-sectional correlational observational study investigated the relationship between maximal isokinetic shoulder strength in both concentric and eccentric internal and external rotations, segmental mass, grip strength, height, and serve speed in junior tennis players. The study complied with ethical research standards and received approval from the Faculty of Physical Education and Sport Ethics Committee, Charles University (No. 243/2020) on 3 November 2020. All procedures were conducted following the Declaration of Helsinki.

### 2.2. Participants

This study involved fifteen male junior players (mean ± SD: age 15.9 ± 0.9 years; height: 180.1 ± 7.2 cm; body mass: 66.1 ± 5.7 kg; body fat: 17.7 ± 2.2%), selected based on their rankings on the junior tennis circuit. The participants in this study were nationally ranked junior tennis players. At the time of data collection, all participants had been training regularly for approximately ten years in a tennis club affiliated with a youth sports center. They trained five times per week, with sessions focused on technical, tactical, and physical preparation. The players regularly competed in national tournaments and International Tennis Federation junior competitions. All participants were familiar with the serve, grip strength, and isokinetic testing protocols. Informed consent was obtained from the legal guardians of all participants after explaining this study’s purpose and procedures. All participants were healthy and free from any traumatic injuries affecting performance or musculoskeletal injuries occurring within three months prior to the start of the study.

### 2.3. Procedure

The testing protocol consisted of two sessions (Figure 1). In the first session, participants initially underwent body composition assessment using dual-energy X-ray absorptiometry (DXA) to determine segmental mass, including arm, trunk, and leg mass. Following the DXA scan, participants completed a warm-up to prepare for grip strength testing. Grip strength was then assessed using a digital hand dynamometer, with participants performing three maximal-effort trials for each hand, and the highest value was recorded. After completing grip strength testing, participants engaged in a warm-up to prepare for isokinetic testing. Isokinetic shoulder strength was measured using an isokinetic dynamometer at two angular velocities: 210°/s and 300°/s. Participants executed three maximal-effort repetitions for internal and external shoulder rotations at each velocity, with standardized rest intervals between tests. The order of testing (angular velocity and internal vs. external rotation) was randomized to avoid potential order effects. The session was conducted under standardized indoor conditions at a controlled temperature of 22 ± 1 °C. Session two was conducted 24 h after the first session and focused on measuring serve speed. After completing a warm-up to prepare for serve speed testing, participants performed a series of 20 maximal-effort flat serves aimed at designated service boxes. Serve speed was recorded using a radar gun, and the session was conducted under the same standardized indoor conditions as the first session. Both sessions were designed to ensure safety, consistency, and reliable data collection.

### 2.4. Instruments

The methods and instruments used for measuring body composition, grip strength, isokinetic strength of the shoulder rotators, and tennis serve speed are detailed in this section in the order they were conducted.

#### 2.4.1. Dual-Energy X-Ray Absorptiometry

Participants underwent body composition analysis using dual-energy X-ray absorptiometry to assess the total mass of specific body segments. Participants removed all metallic and inorganic items, such as jewelry and belts, to prevent imaging artifacts before the scans. They were positioned supine on the scanning table and instructed to remain motionless to ensure consistent and reliable measurements [25]. The DXA scans were conducted using a narrowed fan-beam system (Lunar Prodigy; GE Healthcare, Madison, WI, USA) and analyzed with GE Encore 12.30 software. The system was calibrated daily with a standard phantom to ensure accuracy [26]. Total mass values were extracted for the serving arm, dominant leg, and trunk.

#### 2.4.2. Grip Strength

Before grip strength testing, participants completed a 5 min warm-up consisting of hand and forearm activation exercises, including wrist flexion and extension with a light resistance band (2 sets of 8 reps), squeezing a stress ball (2 sets of 8 reps per hand), and progressive maximal grip contractions using a dynamometer (1 set of 5 reps with increasing force). Grip strength was assessed using a calibrated hand dynamometer (Model GRIP-D, T.K.K. 5401, Takei Scientific Instruments Co., Ltd., Niigata, Japan). Participants performed the test in a seated position with their elbow flexed at 90 degrees and their forearm in a neutral position, in accordance with standardized testing protocols [27]. The dominant hand was used for all measurements. Each participant was instructed to squeeze the dynamometer with maximum force for 3 s, ensuring consistent effort across trials. Three trials were conducted, with a 30 s rest period between attempts to minimize fatigue. The highest recorded value was used for analysis. The dynamometer was calibrated before testing to ensure accuracy.

#### 2.4.3. Isokinetic Strength Testing of the Shoulder

All isokinetic strength tests were performed using a standard isokinetic dynamometer (Humac Norm; CSMi, Stoughton, MA, USA) measuring internal and external rotations of the dominant (serving) shoulder. Tests were conducted at angular velocities of 210°/s and 300°/s to evaluate peak torque during concentric and eccentric muscle actions [28]. These speeds were chosen because predominant high-speed contractions are prevalent in tennis [29,30,31]. The isokinetic dynamometer was calibrated at the beginning of the test day according to the manufacturer’s guidelines to ensure accurate and reliable measurements.

Prior to isokinetic testing of the shoulder rotators, participants completed a comprehensive warm-up to promote muscle readiness and reduce injury risk. This protocol included general aerobic exercises (5 min light jogging with leg swings, arm circles, and trunk rotations) followed by shoulder-specific movements (5 min low-resistance shoulder internal rotations, 3 sets of 10 reps, and external rotations, 3 sets of 10 reps, using resistance bands) to prepare the muscles for the demands of isokinetic evaluation [32]. Participants were positioned supine with the shoulder joint aligned with the dynamometer’s axis of rotation, with the arm abducted to 90° and the elbow flexed at 90°. The shoulder was stabilized with a custom-made humeral support. This position offered stability and closely resembled the serving motion’s arm position [29]. The testing range of motion was set to 90% of each participant’s maximum external rotation and 65% of their maximum internal rotation, ensuring safety during testing while mimicking the functional shoulder movements typically used during the serve. Each participant performed three repetitions of maximum voluntary contractions for both concentric and eccentric muscle actions at each angular velocity. A rest period of 90 s was provided between tests at different velocities to minimize fatigue. To ensure safety during testing, all participants were supervised by a physiotherapist who monitored proper technique and alignment throughout the protocol. Participants were instructed to exert maximum effort during each repetition while maintaining smooth and controlled movements. They were explicitly directed to avoid compensatory movements such as excessive trunk rotation or shoulder elevation, which could compromise measurement validity and participant safety. The order of testing (angular velocity and internal vs. external rotation) was randomized to avoid potential order effects.

#### 2.4.4. Serve Testing Protocol

Following the standardized 15 min warm-up, which included 5 min of light jogging and dynamic exercises such as calf raises, hip hinges, lunges, squats, and hopping, as well as 10 min of specific shoulder activation exercises (external and internal rotations at 90-degree abduction, 3 sets of 12 reps; scapular retractions, 3 sets of 12 reps; and diagonal shoulder patterns, 3 sets of 10 reps per arm) using resistance bands to warm up the rotator cuff muscles and scapular stabilizers, participants proceeded to the serve testing protocol. The shoulder warm-up focused on internal and external rotations and scapular stabilization to ensure optimal preparation and reduce the risk of injury. The session emphasized safety, consistency, and maximal effort, with participants instructed to rest briefly between serves to minimize fatigue.

Before the main testing, participants underwent a familiarization process in which they performed five practice serves aimed at the designated target area to adjust their technique and acclimate to the testing environment. Subsequently, each participant then performed 20 maximal-effort flat serves aimed at the deuce side (right-handed server perspective). The serves were executed in four sets of five serves, with a 30 s rest between each set to minimize fatigue. The focus of this study was on first serves, and therefore, the protocol was not designed to replicate the match conditions. The target area was a rectangle measuring 1 × 2 m, positioned 1 m from the centerline and extending 2 m from the service line towards the net on the deuce side of the tennis court, visibly marked with white tape [3,7]. Serve speed was measured using a radar gun (Stalker Pro II, Applied Concepts Inc., Richardson, TX, USA) placed in front of the server and aligned with the ball’s flight path to ensure accurate speed detection while maintaining safety protocols.

A three-person team facilitated testing to ensure data accuracy and compliance with protocols. One researcher recorded the speed of each serve and communicated the values, while another team member logged valid attempts and documented the results. Only serves that landed within the designated target area and adhered to the prescribed rules were considered valid for analysis. The third team member provided tennis balls to the participants, monitored adherence to serving rules, and ensured that serves were performed from behind the baseline in accordance with official tennis regulations.

### 2.5. Data Collection

The primary variables measured included peak torque values during concentric and eccentric shoulder rotations at angular velocities of 210°/s and 300°/s, assessed using an isokinetic dynamometer. Serve speed was recorded as the mean of the three fastest valid serves, measured with a radar gun. Body composition, including segmental mass and lean mass (arm, trunk, and leg), was assessed using DXA. Grip strength was measured using a digital hand dynamometer, with the highest value from three trials for each hand recorded.

### 2.6. Statistical Analysis

Statistical analyses were performed using IBM SPSS Statistics for Windows (version 25.0, IBM Corp., Armonk, NY, USA) with the PROCESS macro for mediation and moderation analysis (developed by Andrew F. Hayes), and Microsoft Excel 2019 (version 2312, Microsoft Corp., Redmond, WA, USA). Data were tested for normality using the Shapiro–Wilk test, as mediation and moderation analyses were conducted, which typically rely on parametric statistical assumptions. Spearman’s correlation coefficient was used to analyze the relationships among isokinetic shoulder strength (Nm), segmental mass and lean mass (kg), height (cm), grip strength (kg), and serve speed (km/h). The correlations and mediation results were visualized through scatter plots with regression lines to illustrate relationships. A mediation analysis was conducted to evaluate whether the relationship between isokinetic shoulder strength and serve speed was mediated by segmental mass. This analysis followed the Baron and Kenny approach [33] detailed by Hayes [34], testing three relationships. In accordance with this procedure, we adapted this framework into these three relationships for our study: (1) shoulder strength and segmental mass, (2) shoulder strength and serve speed, and (3) the combined effects of shoulder strength, segmental mass, and serve speed. A bootstrap analysis (1000 iterations) was performed to estimate the indirect effect and its confidence intervals. A moderation analysis was also conducted to assess whether height moderates the relationship between shoulder strength and serve speed. Interaction terms (e.g., height × shoulder strength) were included.

A sensitivity analysis was conducted using G*Power (version 3.1.9.6) to determine our sample’s minimum detectable effect size for the correlation coefficient [35]. This study used a two-tailed test with an alpha error probability set at 0.05 and a power (1 − β) of 0.80 (80%). The computed minimum detectable correlation coefficient was r = 0.58. Consequently, the strength of correlations was interpreted as weak (<0.58), moderate (0.58–0.79), or strong (≥0.80), based on an adjusted classification adapted from Cohen [36]. This approach ensured that our sample size was sufficient to detect statistically significant correlations.

## 3. Results

Table 1 presents the descriptive statistics for the performance metrics of isokinetic net moment, tennis serve speed, grip strength, and segment masses. The results indicate a range of means and variability, with internal shoulder rotation eccentrically at 210°/s showing the highest standard deviation (SD = 8.2 Nm) and arm mass the lowest (SD = 0.4 kg). Confidence intervals for the means are relatively narrow across variables, indicating consistent measurements among participants. The Shapiro–Wilk test results demonstrate that most variables do not significantly deviate from normality (*p* > 0.05), except for external shoulder rotation eccentrically at 210°/s (*p* = 0.01), which indicates a non-normal distribution. This result and the small number of participants suggest that non-parametric methods may be more appropriate.

### 3.1. Correlation Analysis Between Serve Speed and Performance Metrics or Segment Mass

Spearman’s correlation coefficient was used to determine the associations between serve speed and selected variables (Table 2). Moderate positive correlations were revealed between serve speed and internal concentric shoulder rotation at 210°/s, external concentric shoulder rotation at 210°/s, external concentric shoulder rotation at 300°/s, arm mass, trunk mass, leg mass, and height. Additionally, the lean mass of the arm and body mass showed strong positive associations, highlighting the role of muscle mass in serve performance. Spearman’s correlation for internal eccentric shoulder rotation at 210°/s and external eccentric shoulder rotation at 210°/s revealed weak positive but non-significant relationships. Weak correlations were also observed for grip strength.

### 3.2. Graphical Representation of Findings

Several key findings emerged from the correlation analysis. Figure 2 illustrates the correlation between serve speed and lean arm mass (Figure 2a) and between serve speed and player height (Figure 2b). The regression lines suggest positive relationships, indicating that greater arm mass and height are associated with faster serve speeds. The stronger relationship observed between serve speed and lean arm mass (R^2^ = 0.734, R^2^_adjust_ = 0.713) compared to height (R^2^ = 0.359, R^2^_adjust_ = 0.309) highlights the importance of arm mass for serve performance. However, this finding is closely tied to player height, as it is logical that taller players have longer arms, which naturally results in greater arm mass. Therefore, the combined effect of height and lean arm mass likely contributes to the observed increase in serve speed, as longer limbs provide a mechanical advantage for generating higher velocities.

Figure 3 illustrates the relationship between serve speed and isokinetic internal shoulder rotation (Figure 3a) and external shoulder rotation (Figure 3b), both measured concentrically at 210°/s. The regression lines in both graphs show positive relationships, suggesting that greater isokinetic strength in the shoulder rotators is associated with faster serve speeds. The relationship between serve speed and external rotation strength (R^2^ = 0.507, R^2^_adjust_ = 0.469) was similar to internal rotation strength (R^2^ = 0.495, R^2^_adjust_ = 0.456). This highlights the important role of external rotators in stabilizing and decelerating the arm during the serving motion, while internal rotators contribute to the acceleration phase.

### 3.3. Mediation Analysis

The mediation analysis assessed whether lean arm mass mediates the relationship between isokinetic concentric shoulder strength and serve speed. The first regression model showed that isokinetic internal shoulder strength concentrically at 210°/s positively predicted lean arm mass (β = 0.0197, *p* = 0.079), although the relationship was not statistically significant. The second regression confirmed that shoulder strength significantly predicted serve speed (β = 1.1192, *p* = 0.003). The third regression, which included both shoulder strength and lean arm mass, demonstrated that both predictors contributed significantly to serve speed, with shoulder strength (β = 0.6170, *p* = 0.010) and lean arm mass (β = 25.5472, *p* < 0.001). Bootstrap analysis (1000 iterations) estimated the indirect effect of lean arm mass as 0.502, with a 95% confidence interval ranging from −0.1564 to 1.1450. These results suggest partial mediation, but the indirect effect was not statistically significant, indicating that lean arm mass may contribute to the relationship between shoulder strength and serve speed, though this effect could not be confirmed with sufficient confidence.

### 3.4. Moderation Analysis

The moderation analysis tested whether height moderated the relationship between isokinetic concentric shoulder strength (210°/s) and serve speed. Interaction terms (height × shoulder strength) were included in regression models, but none of the terms were statistically significant (*p* > 0.05). Therefore, the results do not support height as a significant moderator in the relationship between shoulder strength and serve speed.

### 3.5. Summary of Findings

A moderate positive correlation was observed between serve speed and concentric external shoulder rotation strength at 210°/s (ρ = 0.71, *p* = 0.003). Concentric internal shoulder rotation strength at 210°/s also showed a significant positive correlation with serve speed, though to a slightly lesser extent (ρ = 0.61, *p* = 0.016). A strong positive correlation was found between height and body mass (ρ = 0.84, *p* = 0.0001), as well as between height and lean body mass components such as lean arm mass (ρ = 0.85, *p* = 0.001). However, no significant correlation was found between height and concentric internal shoulder rotation strength (ρ = 0.01, *p* = 0.985), and the relationship between height and concentric external shoulder rotation strength was weak and not statistically significant (ρ = 0.21, *p* = 0.450).

The mediation analysis indicated that lean arm mass partially mediates the relationship between concentric internal shoulder rotation strength and serve speed. The indirect effect of lean arm mass on serve speed was estimated to be 0.502 (95% CI: −0.156 to 1.145), suggesting that lean arm mass contributes significantly to the relationship, although the mediation was not statistically significant. Additionally, the moderation analysis showed that height does not significantly moderate the relationship between shoulder strength and serve speed, as interaction terms were not statistically significant (*p* > 0.05).

These findings indicate that while height and related mass variables contribute to serve speed through structural advantages (longer levers and greater arm mass), isokinetic shoulder strength, particularly concentric external or internal rotation, remains an independent predictor of serve speed. Furthermore, lean arm mass may enhance the relationship between shoulder strength and serve speed, though the effect appears limited in this sample.

## 4. Discussion

The tennis serve is widely regarded as one of the sport’s most physically demanding skills, requiring precise coordination, technical mastery, and substantial muscular strength across multiple body segments [1,2]. Previous studies have demonstrated that concentric and eccentric strength significantly impact racquet head speed among elite players, with stronger shoulders contributing to higher ball velocities and greater serve efficiency [5,12]. Our findings confirm that shoulder strength, particularly in the external rotators during concentric actions, is pivotal in generating the high velocities needed for effective serves in junior tennis players. These findings underscore the complementary roles of these muscle groups in optimizing serve performance, reinforcing the need for balanced strength development [14,37] and confirming the first part of our hypothesis regarding the relationship between shoulder strength and serve speed in junior tennis players.

In addition to shoulder strength, anthropometric factors such as height and body mass emerged as critical determinants of serve speed. The relationship between height and serve velocity in this study corroborates with findings by Bonato et al. [5] and Sanchez-Pay et al. [24], who identified the relationship between height or body mass and serve velocity in professional tennis players. Taller players may initially benefit from biomechanical advantages such as longer levers, which allow for higher racquet velocities [38]. However, height was moderately associated with serve speed but showed no significant relationship with shoulder strength. Instead, the relationship between height and performance appears to be mediated by body mass, particularly lean arm mass, which was associated with height and serve speed. Mediation analysis suggested that lean arm mass may contribute to the biomechanical advantages of shoulder strength, but the effect was not statistically significant. This finding indicates that while taller players benefit from structural advantages such as longer levers and greater arm mass, shoulder strength remains an independent predictor of serve speed in junior tennis players. Contrary to our second part of the hypothesis, height did not significantly moderate the relationship between shoulder strength and serve speed, as interaction terms involving height were not statistically significant. Overall, our findings suggest that while structural advantages associated with height and lean body mass may enhance serve performance, the influence of isokinetic shoulder strength on serve speed operates independently of height.

### 4.1. Concentric and Eccentric Strength in Relation to Serve Speed in Junior Tennis Players

One of the key factors influencing serve performance is shoulder rotator strength, particularly the balance between concentric and eccentric strength [4,9,16,17,23,31,39,40,41]. Eccentric strength is often emphasized due to its role in deceleration and injury prevention, especially at higher racquet velocities, as seen in elite-level tennis athletes with extensive training adaptations [13]. Johansson et al. [40] found that eccentric external rotation strength and intermuscular ratios (eccER/IR) are lower in adolescent players competing at regional levels compared to national-level players, highlighting the importance of eccentric strength as tennis athletes progress. However, while eccentric strength is emphasized in studies such as Johansson et al. [40] for its critical role in deceleration and movement control, our results suggest that concentric strength plays a more prominent role in junior players’ serve performance. Specifically, our findings reveal that concentric shoulder strength in junior players is strongly associated with serve speed, particularly at higher testing velocities, which aligns with the results of Olmez et al. [31]. Therefore, training programs should prioritize developing concentric power while maintaining sufficient eccentric strength to ensure shoulder stability and long-term injury prevention.

Differences in methodological approaches may partly explain the varying conclusions regarding the roles of concentric and eccentric strength in serve performance, as outlined by Ellenbecker [30]. For example, Johansson et al. [40] focused on isometric and eccentric strength ratios normalized to body mass, whereas our study measured maximal concentric and eccentric strength at high velocities using isokinetic protocols. This suggests that concentric strength may be more relevant to explosive movements like the serve, particularly in younger athletes who have not yet fully developed eccentric control mechanisms seen in elite players. Given these findings, future research should aim to standardize testing protocols to better determine the specific contributions of concentric and eccentric strength in different stages of player development.

### 4.2. Height and Mass in Relation to Shoulder Strength and Tennis Serve Speed

In addition to shoulder strength, anthropometric factors such as height and body mass emerged as critical determinants of serve speed. The relationship between height and serve velocity in this study corroborates with findings by Bonato et al. [5] and Sanchez-Pay et al. [24], who identified the relationship between height or body mass and serve velocity in professional tennis players. Taller players may initially benefit from biomechanical advantages such as longer levers, which allow for higher racquet velocities [38]. However, height did not correlate significantly with shoulder strength, indicating that its contribution to serve speed operates independently of muscular factors. Lean arm mass, strongly associated with height and serve speed, was identified as a key variable linking anthropometric characteristics to performance, consistent with previous research demonstrating the importance of limb mass in force generation during high-speed movements [9].

Our findings align with these perspectives, as the association between height and serve speed in our junior players likely reflects the early-stage exploitation of these biomechanical advantages. Over time, as players mature and undergo strength training, the contribution of neuromuscular factors may become increasingly significant. This underscores the importance of integrating physical development and skill refinement into training programs, particularly during the critical developmental stages. Such an integrated approach could help players maximize their biomechanical potential while addressing the physical demands of the sport.

### 4.3. Integration of the Study Findings

From a practical perspective, the balance between internal and external rotators must be carefully developed to optimize serve performance. While external rotators play a key role in stabilizing the shoulder and supporting deceleration, internal rotators are the primary drivers of arm acceleration during the serve [29,31,32]. Training programs for junior players should include exercises targeting concentric strength in both muscle groups, such as resisted shoulder rotations. This aligns with the findings demonstrating that targeted concentric and eccentric shoulder strength training can improve serve velocity in tennis players [28]. Additionally, the strong relationship between lean arm mass and serve speed suggests that strength training to increase muscle mass in the upper extremities could benefit players with less favorable anthropometric profiles. As height did not moderate the relationship between shoulder strength and serve speed, this indicates that lean arm mass mediates the biomechanical advantages. Taller players, who naturally benefit from structural advantages such as longer levers, may need to focus on maximizing their biomechanical potential through technique optimization and stabilizing shoulder strength. In contrast, shorter players may rely more heavily on acceleration strength to offset the disadvantages of shorter levers, highlighting the need for individualized training programs. Future research could explore the potential for asymmetry in shoulder rotator strength in junior tennis players. Asymmetry could influence energy transfer efficiency during the serve, contributing to the development of individualized rehabilitation and training programs. Investigating the longitudinal effects of strength training, particularly at high angular velocities, may provide further insights into optimizing serve performance in junior tennis players.

### 4.4. Limitations

This study has several limitations that should be considered. First, the small sample size (15 players) may limit the generalizability of the findings to a broader population of junior tennis players. Future research with more extensive and diverse samples, including female players and players of different ages and skill levels, must confirm these findings. Second, the cross-sectional design prevents us from drawing causal inferences about the relationship between shoulder strength, anthropometric factors, and serve velocity. Longitudinal studies are needed to examine how changes in these factors over time affect serve performance. Third, while isokinetic testing provides valuable insights into shoulder strength, it does not fully replicate the dynamic and explosive nature of serving in a match environment. Lastly, this study did not account for all potential factors influencing serve speed, such as serving technique, playing surface, and individual player characteristics like flexibility, coordination, or neuromuscular control. Despite these limitations, this study provides valuable insights into the factors contributing to serve performance in junior tennis players.

## 5. Conclusions

This study underscores the critical factors influencing serve performance in junior tennis players by integrating shoulder strength and anthropometric characteristics. Our findings highlight concentric shoulder strength’s contribution, particularly in internal and external rotators, to tennis serve speed. While height and lean arm mass offer structural and muscular advantages, shoulder strength emerges as an independent and trainable variable that directly contributes to serve performance. The mediating role of lean arm mass further emphasizes the importance of muscular development in optimizing serve speed, particularly for players with less favorable anthropometric profiles. Our results also reinforce the necessity of balanced strength development in internal and external rotators, specifically focusing on acceleration strength training to mimic the demands of competitive tennis. Coaches should consider these findings when designing targeted training programs, emphasizing individualized approaches that leverage each player’s unique anthropometric characteristics. By incorporating strength and anthropometric considerations into training plans, coaches can enhance serve performance in junior tennis players more effectively.

## Figures and Tables

**Figure 1 jfmk-10-00057-f001:**
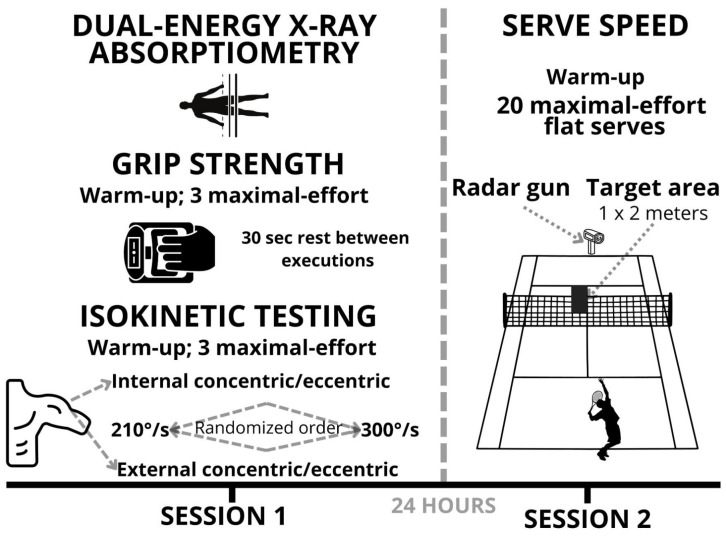
Overview of the testing protocol.

**Figure 2 jfmk-10-00057-f002:**
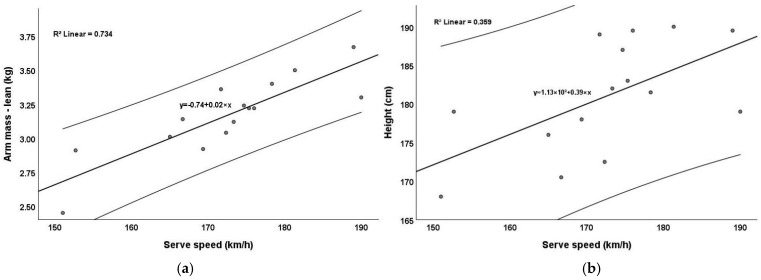
Correlation between the serve speed and lean arm mass or height: (**a**) lean arm mass and serve speed; (**b**) height and serve speed.

**Figure 3 jfmk-10-00057-f003:**
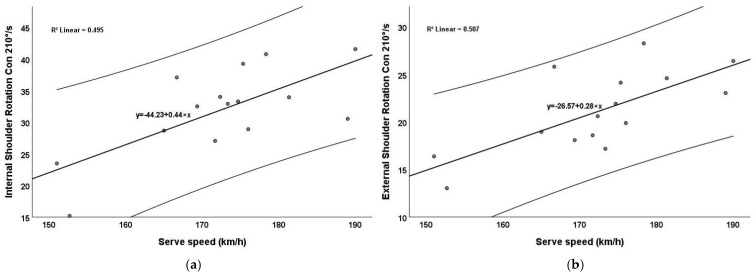
Correlation between the serve speed and shoulder rotation: (**a**) internal shoulder rotation concentric (210°/s) and serve speed; (**b**) external shoulder rotation concentric (210°/s) and serve speed.

**Table 1 jfmk-10-00057-t001:** The performance of isokinetic net moment, serve speed, grip strength, and description statistics of segment mass.

Variables	Mean	SD	CI Lower	CI Upper	SW
Internal Shoulder Rotation Con (210°/s, Nm)	32.0	6.9	28.2	35.8	0.42
External Shoulder Rotation Con (210°/s, Nm)	21.1	4.2	18.8	23.5	0.98
Internal Shoulder Rotation Ecc (210°/s, Nm)	38.6	8.2	34	43.1	0.44
External Shoulder Rotation Ecc (210°/s, Nm)	32.7	7.6	28.5	36.9	0.01
Internal Shoulder Rotation Ecc (300°/s, Nm)	33.5	5.6	30.4	36.6	0.28
External Shoulder Rotation Ecc (300°/s, Nm)	30.1	5.0	27.3	32.9	0.64
Internal Shoulder Rotation Con (300°/s, Nm)	29.1	6.7	25.4	32.8	1.00
External Shoulder Rotation Con (300°/s, Nm)	20.9	3.7	18.9	22.9	0.99
Serve Speed (km/h)	172.4	10.9	166.4	178.5	0.41
Arm Mass (kg)	3.8	0.4	3.6	4	0.96
Arm Mass—lean (kg)	3.2	0.29	3	3.3	0.65
Leg Mass (kg)	11.5	1.4	10.7	12.3	0.55
Leg Mass—lean (kg)	9.3	1.2	8.7	10	0.63
Trunk Mass (kg)	31.9	3.0	30.2	33.6	0.45
Trunk Mass—lean (kg)	26.9	2.8	25.4	28.5	0.95
Body mass (kg)	66.1	5.7	63	69.3	0.9
Body mass—lean (kg)	54.8	5.1	52	57.6	0.96
Grip Strength (kg)	39.7	6.4	36.2	43.3	0.95

Abbreviations: CI—95% confidence interval; SD—standard deviation; SW—Shapiro–Wilk test; Con—concentric; Ecc—eccentric.

**Table 2 jfmk-10-00057-t002:** Spearman’s coefficients between serve speed and performance metrics or anthropometric variables.

Variables	ρ	*p*-Values	Interpretation
Internal Shoulder Rotation Con (210°/s)	0.61 *	0.016	Moderate positive
External Shoulder Rotation Con (210°/s)	0.71 *	0.003	Moderate positive
Internal Shoulder Rotation Ecc (210°/s)	0.41	0.128	Weak positive (ns)
External Shoulder Rotation Ecc (210°/s)	0.39	0.156	Weak positive (ns)
Internal Shoulder Rotation Ecc (300°/s)	0.35	0.196	Weak positive (ns)
External Shoulder Rotation Ecc (300°/s)	0.27	0.328	Weak positive (ns)
Internal Shoulder Rotation Con (300°/s)	0.56	0.030	Weak positive
External Shoulder Rotation Con (300°/s)	0.65 *	0.008	Moderate positive
Arm Mass (kg)	0.77 *	0.001	Moderate positive
Arm Mass—lean (kg)	0.85 *	0.001	Strong positive
Leg Mass (kg)	0.62 *	0.014	Moderate positive
Leg Mass—lean (kg)	0.66 *	0.002	Moderate positive
Trunk Mass (kg)	0.72 *	0.003	Moderate positive
Trunk Mass—lean (kg)	0.60 *	0.017	Moderate positive
Grip Strength (kg)	0.46	0.084	Weak positive (ns)
Height (cm)	0.68 *	0.006	Moderate positive
Body mass (kg)	0.81 *	0.001	Strong positive
Body mass—lean (kg)	0.73 *	0.002	Moderate positive

Abbreviations: ρ—Spearman’s correlation coefficient; ns—not significant (*p* > 0.05). * ρ ≥ 0.58; Con—concentric; Ecc—eccentric.

## Data Availability

The associated dataset for all performed analyses is available at the Open Science Framework [OSF] repository URL: https://osf.io/hb268/?view_only=a1198180313340a5a1ef6ac009c633d9 (accessed on 3 January 2025).

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
