# Peer review of "Tennis Serve Speed in Relation to Isokinetic Shoulder Strength, Height, and Segmental Body Mass in Junior Players"

_jfmk, 2025, doi:10.3390/jfmk10010057_

Round 1
Reviewer 1 Report
Comments and Suggestions for Authors
First of all, I would like to thank you for the opportunity to review this article. This research paper shows an investigation into the importance of various physical and biomechanical aspects of the tennis serve. A very interesting topic, because the serve is becoming more and more relevant in the game, being the initial stroke with which the ball is put into play and which allows to start dominating a point.
However, I would like to make a number of recommendations in order for this work to be published:
1. It would be important that the objective of the research is well defined.
2. In the procedure, it is important that the authors detail the warm-up performed with the players, as one of the main characteristics is that this same research can be replicated.
3. Which ranking are we talking about? International ranking or national ranking? It would be good to know in order to know the level of the players.
4. Did the players serve 20 times in total, without a break, 10 to each side? It would be good to corroborate all this data. In addition, it would be advisable to do it in the same conditions in which the game is played, i.e. the players have had a rest time similar to the points during the course of a match.
5. A section on data analysis is missing. Statistical tests cannot be described in the instruments section.
6. Why was Spearman's correlation test performed and was the sample tested for normality? To be specified in the text.
Finally, I would like to congratulate him because the sections on results, discussion and conclusions are very clear and perfectly justify what the authors want.
Author Response
Comments 1: It would be important that the objective of the research is well defined.
Response 1: Thank you for your valuable suggestion. We have revised the introduction's final paragraph to define our study's objective clearly. The revised section explicitly states that the study aims to determine the correlations between serve speed and isokinetic concentric and eccentric shoulder strength, body mass, segmental masses, height, and grip strength in junior tennis players. Additionally, we have refined the hypotheses to ensure that they align with the correlational nature of our research. Additionally, based on the recommendations from other reviewers, we have also revised and shortened the introduction to improve clarity and readability, ensuring that key concepts are presented concisely while maintaining the necessary depth of information.
Comments 2: In the procedure, it is important that the authors detail the warm-up performed with the players, as one of the main characteristics is that this same research can be replicated.
Response 2: Thank you for your suggestion regarding the structure of the Procedure section. Based on your feedback, we removed the brief descriptions of the warm-up routines from Section 2.2 (Procedure) and provided detailed descriptions of the warm-up protocols within the specific testing sections. The warm-up procedures are now integrated into Sections 2.4.2 (Grip Strength Testing), 2.4.3 (Isokinetic Strength Testing), and 2.4.4 (Serve Speed Testing) to ensure clarity and enhance reproducibility.
Comments 3: Which ranking are we talking about? International ranking or national ranking? It would be good to know in order to know the level of the players.
Response 3: Thank you for your suggestion regarding the ranking level of the participants. We have now explicitly stated in the Participants section that all players were nationally ranked junior tennis players.
Comments 4: Did the players serve 20 times in total, without a break, 10 to each side? It would be good to corroborate all this data. In addition, it would be advisable to do it in the same conditions in which the game is played, i.e. the players have had a rest time similar to the points during the course of a match.
Response 4: Thank you for your question regarding the serve testing protocol. In the Methods section, we have clarified that participants performed a total of 20 serves, all directed to the deuce side, with four sets of five serves, incorporating a 30-second rest between each set to minimize fatigue. Additionally, we have explicitly stated that the protocol focused on first serves only, which is why it did not fully replicate match conditions. This approach was chosen to ensure consistency and control over the serve speed measurements across all participants.
Comments 5: A section on data analysis is missing. Statistical tests cannot be described in the instruments section.
Response 5: Thank you for this request. We separated it into two sections: data collection and statistical analysis.
Comments 6: Why was Spearman's correlation test performed and was the sample tested for normality? To be specified in the text.
Response 6: Thank you for your question regarding the normality testing despite using Spearman’s correlation. Normality was assessed using the Shapiro-Wilk test because the study conducted mediation and moderation analyses, which typically rely on parametric statistical assumptions. While Spearman’s correlation was chosen due to the small sample size. We explicitly added this explanation into the text in the Statistical analysis section.
Reviewer 2 Report
Comments and Suggestions for Authors
This study aims to assess tennis serve speed in relation to isokinetic shoulder strength, height, and segmental body mass in junior players. This study is intriguing, relevant, and methodologically well-executed. However, some improvements are needed before publication, particularly in the discussion of the results.
Introduction:
1. Overall, the introduction is long and hard to read. It is relevant and thorough; however, it should be better written to be easy to read and understand. Try to be “on the point” instead of overexplaining things.
2. Lines 100-105, as a significance of the study, should be added after the aims. Or in the discussion/conclusions. Mainly because the authors are referring to the methodology, which has yet to be explained.
3. Please be aware of how you hypothesise. Correlation is not causation. Therefore, the hypothesis should only be restricted to correlations.
Methods:
1. Ethical considerations:
a. Please add the date of the acceptance
b. Did participants sign the informed consent per the Declaration of Helsinki? Or parents, since these are junior players. Please indicate this in this section.
2. The participant's section should be before the procedures and protocols.
3. Separate 2.4.4. section from the statistical analysis.
4. Also, separate data analysis and statistical analysis. It will be easier to read and understand the methodology.
5. Lines 252 -255 and 262-264 – please add the adequate references.
Results:
Section 3.5. seems unnecessary. Consider deleting it.
Discussion:
1. I have the same problem as with the introduction. It is long and hard to read, with many unnecessary writings.
a. Lines 371-403 – the first paragraph of the discussion should highlight the most important findings. Try to be more concise.
b. Section 4.1. I find it unnecessary to have one entire section only for hypotheses. This can be said in the first paragraph of the discussion. Later, you elaborate on the findings.
c. Same with the chapter 4.2. I’ve never seen this chapter in discussing a scientific paper. This is an integrative part of the discussion and should not be separated.
d. Chapter 4.3. is without references.
2. Basically, this paper does not have a discussion. The only connection to the previous studies and authors' speculations. Try using this approach. Remind us of the finding (in one sentence), then connect that to the earlier studies (also brief), and then elaborate and discuss the finding, citing the relevant references (several sentences). Finally, add your “opinion” on the topic (one sentence)
Author Response
Introduction:
Comments 1: Overall, the introduction is long and hard to read. It is relevant and thorough; however, it should be better written to be easy to read and understand. Try to be “on the point” instead of overexplaining things.
Response 1: We appreciate your feedback regarding the readability of the introduction. Based on your suggestion, we have streamlined several sections by removing redundant explanations while maintaining clarity and logical flow. The revised version now presents key concepts more concisely without compromising the depth of information.
Comments 2: Lines 100-105, as a significance of the study, should be added after the aims. Or in the discussion/conclusions. Mainly because the authors are referring to the methodology, which has yet to be explained.
Response 2: We acknowledge your suggestion and have moved this text, ensuring that the introduction remains focused on presenting the study's background, rationale, and objectives.
Comments 3: Please be aware of how you hypothesise. Correlation is not causation. Therefore, the hypothesis should only be restricted to correlations.
Response 3: Thank you for this important remark. We have carefully revised the hypothesis statements to ensure they strictly refer to correlations rather than causal relationships. The final paragraph now clearly states that we hypothesize positive correlations between serve speed and chosen variables, ensuring scientific phrasing.
Methods:
Comments 1: Ethical considerations:
Comments 1a: Please add the date of the acceptance
Response 1a: Thank you for your request regarding the details of the ethical approval. We have now included the exact date of ethical committee approval in the Methods section and the Institutional Review Board Statement.
Comments 1b: Did participants sign the informed consent per the Declaration of Helsinki? Or parents, since these are junior players. Please indicate this in this section.
Response 1b: Thank you for your comment regarding informed consent. We confirm that all procedures adhered to the Declaration of Helsinki. Informed consent was obtained from all participants, and for minors, consent was provided by their legal guardians in accordance with ethical research standards. This has been explicitly clarified in the „Institutional Review Board Statement“, end of the manuscript.
Comments 2: The participant's section should be before the procedures and protocols.
Response 2: Thank you for your notice. We moved the section Participants before the section Procedure.
Comments 3: Separate 2.4.4. section from the statistical analysis.
Response 3: Thank you for your notice. We moved the statistical analysis into a separate section.
Comments 4: Also, separate data analysis and statistical analysis. It will be easier to read and understand the methodology.
Response 4: Thank you for this request. We separated it into two sections: data collection and statistical analysis.
Comments 5: Lines 252 -255 and 262-264 – please add the adequate references.
Response 5 (Lines 252-255): Thank you for your observation regarding missing references. We have now supplemented the methodological description by adding an additional reference to Andrew F. Hayes' book on mediation and moderation analysis, which provides a detailed explanation of the Baron and Kenny approach. Additionally, we have explicitly stated that the PROCESS macro for SPSS, developed by Andrew F. Hayes, was used to perform mediation and moderation analyses.
Response 5 (Lines 262-264): Thank you for your comment. The sensitivity analysis was conducted using G*Power (we added the reference). This approach was used to ensure that the study had adequate statistical power to detect meaningful correlations. The classification of correlation strength (weak, moderate, strong) was adapted from Cohen (1988) while considering minimum detectable correlation to our sample size.
Results:
Comments 1: Section 3.5. seems unnecessary. Consider deleting it.
Response 1: Thank you for your comment. This text is not inevitable. However, we decided to leave it here. Thank you for understanding.
Discussion:
Comments 1: I have the same problem as with the introduction. It is long and hard to read, with many unnecessary writings.
Response 1: Thank you for your feedback regarding the length and readability of the discussion. Based on your suggestion, we have revised and streamlined the discussion section, removing redundant information and unnecessary details.
Comments 1a: Lines 371-403 – the first paragraph of the discussion should highlight the most important findings. Try to be more concise.
Response 1a: Thank you for your suggestion regarding the first paragraph of the discussion. We have revised this section to be more concise and to clearly highlight the most important findings. The updated paragraph now directly presents the key results, minimizing redundancy and unnecessary details while maintaining clarity.
Comments 1b: Section 4.1. I find it unnecessary to have one entire section only for hypotheses. This can be said in the first paragraph of the discussion. Later, you elaborate on the findings.
Response 1b: Thank you for your suggestion regarding the structure of the discussion. Based on your feedback, we have removed Section 4.1 (Hypotheses) and integrated its content into the introductory part of the discussion.
Comments 1c: Same with the chapter 4.2. I’ve never seen this chapter in discussing a scientific paper. This is an integrative part of the discussion and should not be separated.
Response 1c: Thank you for your feedback regarding Section 4.2. Based on your recommendation, we have restructured this section and integrated its content into the main discussion instead of presenting it separately. We followed the suggested approach by ensuring a logical flow where we first outline the problem, connect it with relevant literature, present our findings, and conclude with our interpretation and recommendations.
Comments 1d: Chapter 4.3. is without references.
Response 1d: Thank you for observing the lack of references in Chapter 4.3. We have now added relevant citations to support the discussion in this section, ensuring that all arguments are properly contextualized within the existing literature. This section focuses on the integration of the findings, in line with the requirements of other reviewers in this scientific journal.
Comments 2: Basically, this paper does not have a discussion. The only connection to the previous studies and authors' speculations. Try using this approach. Remind us of the finding (in one sentence), then connect that to the earlier studies (also brief), and then elaborate and discuss the finding, citing the relevant references (several sentences). Finally, add your “opinion” on the topic (one sentence)
Response 2: Thank you for your detailed feedback regarding the structure of the discussion. Based on your recommendation, we have restructured the discussion section to follow the suggested approach. Each key finding is presented clearly and concisely, followed by a brief connection to relevant literature, an elaboration on our results with supporting citations, and a final interpretation or recommendation. We appreciate your insightful suggestions, which have significantly improved our discussion's clarity, readability, and scientific rigor.
Round 2
Reviewer 2 Report
Comments and Suggestions for Authors
Dear authors,
I appreciate your addressing all the issues and the revisions you made. The manuscript is much improved now.
Best regards